# Investigation of Combustion of KMnO_4_/Zn Pyrotechnic Delay Composition

**DOI:** 10.3390/ma15186406

**Published:** 2022-09-15

**Authors:** Mateusz Polis, Konrad Szydło, Tomasz Jarosz, Marcin Procek, Paweł Skóra, Agnieszka Stolarczyk

**Affiliations:** 1Łukasiewicz Research Network-Institute of Industrial Organic Chemistry, Explosive Techniques Research Group, 42-693 Krupski Młyn, Poland; 2Department of Physical Chemistry and Technology of Polymers, Silesian University of Technology, 44-100 Gliwice, Poland; 3Department of Optoelectronics, Silesian University of Technology, 2 Krzywoustego Str., 44-100 Gliwice, Poland; 4Department of Inorganic Chemistry, Analytical Chemistry and Electrochemistry, Silesian University of Technology, 6 Krzywoustego Str., 44-100 Gliwice, Poland

**Keywords:** delay compositions, pyrotechnic, lead free PDC, high-energetic materials, detonators

## Abstract

This article reports an investigation of the combustion of a binary pyrotechnic delay composition (PDC), consisting of zinc powder as fuel and KMnO_4_ as an oxidising agent, with zinc content ranging from 35 to 70 wt. %. The linear burning rate for delay compositions in the form of pyrotechnic fuses was investigated. Compositions with zinc content between 50 and 70 wt. % yielded burn rates in the range of 13.30–28.05 mm/s. The delay compositions were also tested for their sensitivity to friction and impact, where the compositions showed impact sensitivity in the range from 7.5 to 50 J and were insensitive to friction. Tests in a pressure bomb were carried out to determine the maximum overpressure and pressurisation rate. The thermal properties of the composition were evaluated by thermogravimetric analysis (DTA/TG). The morphology of the combustion products was studied by SEM technique, EDS analyses were used to investigate the element distribution of the post-combustion residues, providing an insight into the phenomena taking place during the combustion of the delay compositions.

## 1. Introduction

Energetic materials have found a broad array of applications, including in aeronautics, civil engineering, as well as in the mining and automotive industries [1]. The use of energetic materials, regardless of specific application, requires precise control over the timing of the energetic event or events, in order to mitigate potential hazards (e.g., constructive interference of seismic waves during blasting operations) and to ensure the desired outcome (e.g., maximising the amount of mined rock with the use of minimal amounts of energetic materials, while achieving the desired rock fragmentation parameters) of the use of energetic materials. Such control is typically exercised with the use of delay detonators that allow discrete time intervals (“delay times”) to be introduced between the sending of the initiating impulse and the initiation of the energetic material charge or charges [2].

Three main classes of delay detonators are currently in use: non-electric [3], electric [4], and electronic [5]. Electronic detonators are the newest class of detonators, offering extremely high precision and repeatability of delay time, due to the delay time being controlled via a microchip. Even so, they have not yet achieved a significant market share due to their high cost in comparison with the other, more traditional classes of detonators and their incompatibility with other types of detonators [6] and susceptibility to overload [7]. Conversely, the delay time in the case of electric and non-electric detonators is achieved through the combustion of a pyrotechnic delay composition (PDC).

The key requirement for PDCs is that they exhibit a linear burning rate that is constant along the length of the PDC column and, ideally, is independent of external conditions. Although in the simplest case, a PDC can consist of a single oxidising agent and single fuel, achieving both the desired performance (sufficient combustion stability and desired linear burning rate value), facilitating PDC manufacture and storage typically requires the use of a variety of additives, such as binders and burning rate modifiers, catalysts, and anti-caking, conserving, waterproofing and anti-corrosion agents [8]. These additives are used in relatively low amounts (e.g., 1–5 wt. % for formulation aides) and typically are either organic substances that are readily decomposed or are substances already present in the environment (e.g., SiO_2_, Fe_2_O_3_).

Conversely, the main components of existing PDC formulations are often substances that are problematic both from a personnel health and from an environmental standpoint, such as heavy metals and their compounds, and chromium(VI) salts [8,9,10,11]. This is not always the case, as recent works have reported some environmentally-neutral PDC formulations, such as W/Fe_3_O_4_ [12], even though such systems may not be competitive economically despite their excellent performance. Consideration for environmental and human health impact in this regard has not yet been fully adopted, as even newly reported PDC formulations contain problematic components, such as perchlorates [13] or chromates [14], even though work is carried out on replacing the latter with alternative materials [15].

In terms of application, currently, two PDC formulations are primarily used to manufacture delay detonators—Si/Pb_3_O_4_ and Sb/KMnO_4_ [16,17]. The former formulation requires personnel to handle micro- or nanometric silicon dust and lead oxide at the manufacturing stage, while contaminating the environment with lead and its salts. Since fulfilling health and environmental safety criteria for this formulation would require replacing both main components (oxidising agent and fuel) of the formulation, it does not appear justified to continue development or use of this PDC formulation.

In turn, for the Sb/KMnO_4_ PDC formulation, potassium permanganate has little environmental impact and has been evaluated as a “green” agent [18]. Antimony, however, is both highly toxic to humans and hazardous environmentally, with both the scope and mechanism of those impacts being extensively documented in literature [19,20,21,22]. Consequently, this PDC formulation may be further developed into a highly desirable “green pyrotechnic delay composition” if an alternative fuel is found to replace antimony.

Zinc is one potential alternative that has virtually no toxicity and environmental impact, either by itself during manufacture or as its compounds after the combustion of the PDC [23,24,25]. It has been studied as a component of PDCs [26,27] utilising various oxidising agents, including in a binary mixture with potassium permanganate. In the investigation of the Zn/KMnO_4_ system reported here, however, the investigation was limited only to heavily compacted (55–165 MPa) formulations and focused primarily on the impact of the share (wt. %) of Zn in the composition on the achieved linear burning rate. No information on the combustion of non-compacted Zn/KMnO_4_ system and no information about pressure build-up during the combustion of this system were, however, reported.

The lack of information about pressure build-up during combustion of a PDC formulation is of particular importance, as PDCs used in detonators are enclosed in metal casing and sufficient pressure build-up during combustion of the PDC can cause the detonator to rupture before the intended delay time has elapsed [8], leading to premature initiation of energetic material charges and misfires. Such issues can potentially compromise the entire blasting operation, adversely affecting the application potential for the “green” Zn/KMnO_4_ PDC formulation.

The aim of this work was to alleviate the above-mentioned issues related to the Zn/KMnO_4_ PDC formulation. Consequently, we have endeavoured to report the information missing from existing works on the subject, as well as to provide a more comprehensive picture of the properties of the Zn/KMnO_4_ PDC formulation. The burning rate, pressure build-up parameters, sensitivity to mechanical stimuli and thermochemical data for this formulation are reported as a function of the share (wt. %) of Zn in the formulation, providing the key information required for the practical application of this PDC formulation in delay detonators.

## 2. Materials and Methods

### 2.1. Materials Characterization

Zinc powder with an average grain size of 3–4 μm (97 wt. % purity), obtained from Selkat S.A (Krakow, Poland), was used as received. Potassium permanganate obtained from LachNer Chemical (Neratovice, Czech Republic) was milled in a circulating crusher for 1 h, followed by sieving to separate grain fractions smaller than 56 μm. Both raw materials materials were investigated via thermogravimetry (DTA/TG).

### 2.2. Preparation of PDC Samples

The components were dried for 12 h at 60 °C, followed by weighing, so as to prepare 30 g of each formulation (Table 1), with the share of zinc being varied in the 35–70 wt. % range. Zn/KMnO_4_ compositions were mixed by brushing the composition through a 75 μm sieve 5 times. The purpose of the sieve-brush-mixing operation was to break up same-particle agglomerates and to facilitate intimate mixing of the formulation components. The prepared compositions were dried for 2 h at 60 °C.

### 2.3. Sensitivity Testing

Compositions were tested for sensitivity to mechanical stimuli. To determine the impact and friction sensitivity of the compositions, the standardised BAM Fallhammer test [28] and Peters’ friction test [29] were employed, respectively.

### 2.4. Preparation of Delay Elements and Grammage Measurement

A lead-antimony alloy (99% of Pb) pipe (50 cm long, 16 × 2.5 mm in diameter) was crimped on one side using a mechanical clamp. The pipe was then dragged trough a set of progressively smaller in diameter eyelets, until the outer diameter of the pipe was equal to 11 mm. The obtained pipe was approximately 100 cm long. Two sections, 50 and 40 cm long, respectively, were cut from the prepared pipe. These elements were then crimped on one side in the same way as before. Lead tubes were filled with 15 g portions of each PDC formulation (Zn1 through Zn7). Each composition was filled into a 50 cm (L-fuse) and into a 40 cm (S-tube) pipe section respectively. The series of delay elements were named according to the key: L + formulation number + L/S (long—50 cm—or short—40 cm—pipe). After that, the PDC-filled pipes were crimped and dragged through another set of eyelets, until their outer diameter was reduced to 6 mm, resulting in pipe sections-pyrotechnic fuses - that were approx. 1 m long. Each of the pyrotechnic fuses was cut into 20 cm long sections. From each fuse, three 5 cm long elements were taken for grammage measurement.

Three 5 cm long sections of each fuse were prepared for testing. After marking, these sections were weighed with an accuracy of 1 mg, and measured with an accuracy of 0.05 mm. Then, by using a sonification bath and wooden drill, the loaded composition was removed. The remaining bodies were washed with distilled water until the characteristic violet-pink color originating from the dissolution of potassium permanganate disappeared. Then, the fuses were dried at the temperature of 60 ± 5 °C for 6 h. After cooling down, the fuses were weighed and then the base casing weight was calculated. Grammage was calculated according to Equation (Equation 1) as follows.
(1)$$G=\frac{\left(m_{1}-m_{2}\right)}{l}$$
where *m_1_* is the mass of the tested fuse, *m_2_* is the mass of the fuse after washing and drying, *l* is the length of the fuse.

### 2.5. Burning Pressure and Burning Rate Investigation

A cylindrical aluminium shell (7 mm high, 9 mm outer diameter, 8 mm inner diameter) equipped with 0.073 mm diameter nichrome resistant wire connected with its bottom part, was filled with a 400 ± 2 mg charge of each tested composition and pressed with 50 N. Each firing set had an electrical resistance in the range of 0.95–1.00 Ω. After preparation, each shell was placed inside a 45 cm^3^ stainless steel constant volume vessel in ambient conditions. The samples were ignited via the hot-wire method, through application of a DC pulse (4.25 A). The internal pressure of the vessel was measured with a PCB Piezoelectronics M102B06 (PCB, Depew, NY, USA), pressure transducers. The transducer signal was recorded using a digital oscilloscope Rigol MS05104 (Rigol, Shanghai, China), with 1 MHz sampling, and analysed with Origin Pro (Origin Lab, Northampton, MA, USA) software.

The fuses were cut into 20 cm long pieces, acting as delay elements. In order to measure the burning rate, four probe holes with a diameter of 1 mm were drilled in each delay element, with the distance between the probe holes being equal to 5 cm. In each case, the distance of the first probe from the initiation point was 3 cm. This distance is aimed at allowing the combustion process to stabilise, so as to exclude the influence of the initiating factor on the measured parameter. Short-circuit type measuring probes, made of insulated copper wire with a diameter of 0.1 mm were threaded through the holes. After the probe was pulled through, it was secured against shifting with insulating tape. The delay elements prepared in this way were dried at a temperature of 60 ± 5 °C for at least 2 h. In the next stage, an aluminum mounting sleeve was clamped on the delay element. Then, a igniter head inside the plastic plug was placed in the sleeve, and clamped. Probes was powered by 5 V voltage source and connected with digital oscilloscope Rigol MS05104, with 100 kHz sampling. The burning rate was calculated based on the response time of successive probes, separated by known distances.

### 2.6. DTA/TG Study

Thermogravimetric studies were performed with a Q-1500 (MOM, Budapest, Hungary) derivatograph, for 200 mg samples, using a heating rate of 10 K/60 s for all samples. The tests were performed in argon or, in the case of zinc, also in an air environment, with a flow rate of 60 dm^3^/h in all cases.

### 2.7. SEM EDS Study

The morphology and chemical composition of the post-combustion residues were investigated using an FEI Inspect S50 (FEI, Hillsboro, OR, USA) scanning electron microscope (SEM) and an X-ray energy dispersive spectrometer with Detector EDS Octane Elect Plus and Analyzer EDAX Z2-i7 (Bruker, Billerica, MS, USA), enabling simultaneous acquisition of microghraphs and Energy Dispersive X-ray (EDS) maps of the investigated samples. In order to ensure the validity of the EDS measurement, the basic SEM operation parameters were that the working distance was 10 mm, the acceleration voltages of the incident electron were 5 kV and 30 kV, the electronic beam spot size was 5 and the current intensity of the incident electronic beam was about 95 μA.

## 3. Results

All tested PDC formulations exhibit friction sensitivity values above 353 N, which allows them to be classified as friction insensitive. The PDC fromulations are, however, sensitive to impact, as tested using a BAM Fallhammer (Table 2).

Impact sensitivity tests showed an initial increase in impact sensitivity with increasing fuel content, followed by a sharp decrease, with maximum sensitivity of Zn5 composition (60% Zn).

### 3.1. Delay-Element Grammage Measurement

The grammage of the tested formulations remained fairly constant, regardless of the formulation used to make the fuse (Table 3). Quite significant weight deviations are related to the limitations of the fuse-drawing method. Despite its advantages and straightforward procedure, this method can lead to small variations in cross-sectional area of the sample and irregular distribution of the composition inside the fuse.

### 3.2. Burning Rate

The burning rate of Zn/KMnO_4_ compositions were measured five times (N = 5) for each PDC formulation (Table 4). Among the tested fuses, only elements containing Zn1 and Zn2 compositions have not been susceptible to ignition from a standard igniter head. This lack of susceptibility to ignition can probably be related to the properties of the compositions themselves and to the geometry and material of the bodies. Literature reports that analogous binary formulations that were compacted achieved burning rates in the range of 1.9–7.3 mm/s [27]. It should be noted that it is recommended to use bodies with an internal diameter greater than 6 mm for compositions burning at a speed lower than 25 mm/s [8].

Zinc, at temperatures just above its melting point is characterised by high viscosity (3.737 Pa·s at 700 K (approx. 423 °C) [30]), which limits diffusion. Therefore, for systems with relatively minor exothermic reaction effects (or with low burning rate) it may not be possible to obtain a stable burning process. In addition, the low melting point of zinc implies the absorption of huge amounts of heat (approximately 7.7 kJ/mol [31]) for its melting at the initial, critical stage of ignition of the composition. The shells of the tested PDCs were made of a lead/antimony alloy. This material exhibits high thermal conductivity and heat capacity, which also drains away the heat evolved during combustion of the PDCs. This could lead to a quick extinction of combustion as a result of heat consumption to increase the casing temperature, thermal losses to the environment, and heat transfer from the pyrotechnic core through the casing material (through the conductive front in the fuse material). The burning rate of the composition increases with increasing fuel content, up to a maximum for the fuel content equal to 65 wt. %. A further increase in the fuel content causes a decrease in the observed burning rate of the formulation. Compositions with a lower fuel content exhibit slightly higher combustion stability. Average burning rates of L3S–L6S (high-density) fuses are slightly higher than those of lower density fuses with corresponding compositions. On the other hand, the average burning rate of the L7S fuse is definitely lower than that of the L7L fuse. It is most likely related to the high zinc content in the composition. Strong compaction of the pyrotechnic core reduces the number of pores, which inhibits gas and liquid diffusion. This in turn invokes the effect of reducing the burning velocity of the composition.

### 3.3. Pressure Parameters

The pressure parameters of Zn/KMnO_4_ compositions were examined with six experiments for each sample (Table 5). The maximum pressure value increases up to a fuel content of 55% and reaches a maximum for this zinc share in the PDC formulation. The increase in the rate of pressurisation is not as rapid, but is distinctive for compositions containing up to 60% of fuel. The maximum pressurisation rate corresponds to a fuel content of 60% in the formulation.

Despite the decreasing oxygen balance, an increase in the measured parameters was observed. This is likely connected with a higher heat of the occurring reactions. The relatively low melting and boiling points of zinc should also be noted, as increasing zinc content will result in increasing the maximum observed pressure due to vaporisation of zinc. Compositions, in which there is a significant excess of fuel, can still generate enough heat to vaporise the fuel. This implies that high pressure parameters can be achieved. The tests reported herein were also performed in air, at ambient pressure, so that the components that did not fully react during combustion could burn out in contact with oxygen from air.

### 3.4. DTA/TG Study

Results of DTA/TG studies of potassium permanganate are shown in Figure 1. The initial mass loss, occurs due loss of moisture. An approx. 12.5% mass loss accompanied by an exothermic effect, with an onset at approximately 250 °C and peak maximum at 290 °C, results from the first step of KMnO_4_ decomposition. The gentle mass loss until 500 °C corresponds to further decomposition, and a release of oxygen. The second step of thermal decomposition of KMnO_4_ with an approx. 1.7% weigh loss, had an onset at approximately 520 °C, with the endothermic peak maximum being observed at approximately 540 °C.

Thermogravimetric DTA/TG of zinc were conducted under air and argon flow (Figure 2). Expectedly, regardless of utilised gas, the melting of Zn was observed at approx. 420 °C, consistent with literature data [32]. Further heating in argon showed a strong endothermic peak accompanied by mass loss, consistent with gradual vaporisation and boiling of zinc. In the case of heating in air, an opposite phenomenon—an exothermic peak accompanied by mass increase—was observed, corresponding to gradual oxidation of Zn to ZnO.

DTA/TG measurements conducted for the Zn/KMnO_4_ PDC formulations (Figure A8) revealed a visible effect of fuel (Zn) content on the thermal behaviour of the PDC formulations. For each DTA/TG plot, the first stage of permanganate decomposition is observed in the initial stage (approximately 270–300 °C) of the experiment. This is accompanied by a mass loss, due to KMnO_4_ reduction to K_2_MnO_4_ [33]. The loss of sample mass is consistent with the calculated value, indicating that the oxygen released at this stage does not undergo a reaction with zinc. This is both due to the inability of oxygen to infiltrate the solid zinc particles, which are covered by an outer shell of ZnO. The successively observed endothermic peak (approximately 420 °C) corresponds to the melting of zinc, as in the case of the thermograms recorded for pure zinc (Figure 2). Further heating leads to a slow increase in the thermal effect of the reactions taking place (at approx. 460 °C) (Table 6), as observed for all investigated PDC formulations. This slow increase is explained by the simultaneous decomposition of the oxidising agent (endothermic) and oxidation of zinc (exothermic). The decrease in the temperature at which the oxidising agent decomposition process occurs may be due to the effect of zinc oxide, with which the fuel particles are coated. The point, at which full ignition of the composition takes place is approximately 505–515 °C (Table 6). It is possible that the interval between these points is related to the ignition induction period, where the endothermic decomposition of the oxidising agent particles suspended in the molten fuel matrix absorbs enough heat to prevent full ignition.

Simultaneously, the slow release of oxygen in the viscous liquid medium, limits the reaction to the immediate vicinity of the oxidising agent molecules. The change in ignition temperature is related to the increasing impact of the features of the fuel on the macroscopic properties of the formulations, increasing the need for the oxygen produced during the decomposition of the oxidising agent to diffuse through the molten fuel and the products of combustion surrounding the oxidising agent particles. Only when the temperature rises above a certain value are the critical conditions exceeded and the PDC fully ignited. These critical conditions are likely tied to the vaporisation of zinc, which induces movement in the liquid zinc matrix, promoting its infiltration by available oxygen.

### 3.5. SEM-EDS Study

Analysis of the SEM micrographs (Figure 3) reveals the presence of well-formed, large, regular crystals that form sintered agglomerates with a clearly layered structure for compositions with low fuel content. As the fuel content increases, the uniformity of the sinter decreases and the number of pores, caverns, and cracks increases noticeably. Spindle-shaped crystals become visible for Zn2 and Zn3 compositions, needle-like crystals become visible for Zn4 compositions and the microstructure becomes less and less crystalline as zinc content is further increased. Simultaneously, an increasing number of partially oxidised spherical fuel particles is observed. The proportion of solidified zinc matrix and the number of unbound sintered particles also increases.

For systems containing small amounts of fuel (up to 45%) on the basis of EDS analyses (Figure 4 and Figure A1, Figure A2, Figure A3, Figure A4, Figure A5, Figure A6 and Figure A7 in Appendix A), a point concentration of manganese atoms, a very uniform distribution of potassium atoms and a concentration of zinc atoms around points of high manganese concentration are observed. This may potentially indicate a limiting process at the fuel–oxidising agent interface, and allows us to speculate on the diffusion of liquid zinc into the oxidising agent crystals, where the reaction is limited by mass transport and the oxide layer that forms in situ. The even distribution of potassium atoms indicates its migration, which is probably related to the formation of intermetallic oxides.

The SEM images taken for the Zn4 and Zn5 compositions indicate high homogenization of all components—possibly an optimal system both kinetically and thermodynamically. The high heat of reaction allowing dynamic mass exchange and melting or evaporation of a significant part of the fuel may be responsible for the high homogenization. The migration of manganese can be associated with the highest values of the maximum overpressure and pressure build-up rate for these compositions. As the fuel content of the compositions increases, the degree of zinc homogenization in the composition increases; this is related to the filling of available spaces by the liquid and resublimating fuel, which is unable to burn due to the negative oxygen balance. In the case of Zn6 and Zn7 compositions, it is possible to notice the re-concentration of manganese atoms in small areas, good homogenization of potassium atoms, and the formation of a “matrix” by zinc. The limited diffusion of manganese may be due to the very high viscosity of the fuel-rich system, and its limited availability with the associated faster consumption. It may also be due to the evaporation of most of the fuel and the occurrence of reactions in a multi-phase system. The results indicate that the reaction for compositions containing up to 60% fuel (Zn5) is limited by the diffusion of zinc into the oxidising agent crystals, but for compositions with higher fuel content, the most likely limiting process is a mechanism limited by the competition between the high proportion of gaseous zinc and the high viscosity of the system formed mainly by liquid zinc.

## 4. Discussion

The pyrotechnic delay composition formulations discussed in this paper exhibit a number of characteristics both favorable and unfavorable from the point of view of their use as delay compositions. Compositions with a fuel content in the range of 50–60% exhibit a relatively stable burning rate, as compared to other PDC systems. However, it should be noted that the repeatability of these parameters is far too low to use the compositions without any need for additives or process optimisation. Although such repeatability issues can be readily overcome, this requires an extensive R&D effort geared towards a set of conditions specific to the design of detonators and delay elements used, rather than the raw material properties of the PDC formulations. The use of the lead-fuse drawing method, despite allowing small batches of delay elements to be readily produced, contributes to increasing such scatter, as it is susceptible to producing uneven density of the compositions along the length of the fuse, fluctuations in the outer or inner diameter and wall thickness of the fuses, as well as hollow zones inside the fuses.

The tested compositions containing 50–70 wt. % of fuel showed the expected sensitivity to the initiating stimulus. This provides basis for the assumption that the compositions in question are capable of transferring ignition, when used in the fire chain of an igniter.

The advantage of the studied compositions is the widespread availability and low price of the raw materials, as well as their minimal impact on human health and the environment, unlike the components of the most commonly used PDCs. The properties of the raw materials, make it technologically simple to process them into a form that allows them to be economically used in the composition of pyrotechnic compositions and using standard apparatus used for the preparation of pyrotechnic compositions.

The less negative standard redox potential of Zn (fuel) relative to the redox potential of the potassium cations present in the oxidising agent prevents the formation of corrosion cells. In the solid state, the contact area between the Zn and KMnO_4_ grains is limited and the formation of an oxide layer on the surface of Zn (fuel) grains further limits the scope of unintended reactions between the two components. The only potential factor that can promote reactions between the two components is the presence of moisture, as KMnO_4_ is readily soluble in water. Due to the fact that the products of KMnO_4_ decomposition are largely insoluble in water, their presence is not expected to adversely influence the stability of the PDC formulation. Consequently, the long-term stability of the PDC formulation is expected to be relatively high, provided that the PDC is stored in dry conditions.

In the case of a simplified model of a stable self-sustaining combustion process, the laminar reaction zone sustains the propagation of the process by heating the adjacent material layer, allowing it to ignite [11]. In the initial stage, reactions typically proceed in the solid phase until enough heat is generated to melt a significant portion of the reagents or to induce decomposition of the oxidising agent on a macroscopic scale.

When considering the effect of burning rate on the mechanism of the combustion process, the evolution of gas from the composition should be taken into account. In the case of the compositions tested in this work, the maximum burning rates are correlated with the rate of pressure build-up. This constitutes a basis for postulating the dominant influence of the gaseous decomposition products driving the mass movement of liquid and solid material responsible for most of the heat transfer. In such a situation, heat transfer is dominated by convection. The porosity of the combustion products visible in the SEM images supports this conclusion. This hypothesis is also supported by the lower burning rates of Zn3 and Zn4 compositions, where a greater contribution of conduction to heat transfer can be expected. Heat removal from the reaction zone by the casing should also be considered. In the case of fast-burning compositions, heat transfer through the casing may be less significant relative to the propagation of the front in the pyrotechnic core. In the case of Zn1 and Zn2 compositions, this may explain the lack of combustion propagation in the pyrotechnic fuse. This is corroborated in literature, as the use of casings with an inner diameter larger than 6 mm for compositions burning at a lower rate than 25 mm/s is recommended [8].

The investigated compositions are susceptible to reaction with atmospheric factors, i.e., moisture. This is related to the properties of potassium permanganate. An alkaline, humid environment leads to the partial reduction of potassium permanganate to MnO_2_. In an acidic environment, reduction of potassium permanganate to MnO is possible [34]. Both of these reactions lead to lowering of the composition’s performance and even deactivation of the PDC. The decomposition of KMnO_4_ in an acidic environment produces Mn^2+^ compounds, which can act as KMnO_4_ decomposition catalysts. Their formation will lead to increase in sensitivity to initiating stimuli and thus, a decrease in safety of operation and use. As such, storage of the PDC should be realised using moisture-resistant and pH neutral vessels.

The tested compositions are insensitive to friction. The sensitivity to impact is strongly dependent on the composition of the compositions. However, these are still values that allow the safe use of products containing these compositions.

The effect of composition density on combustion reaction parameters cannot be fully postulated. The difference in the grammage of the delay elements tested was small, also the differences in the burning rate of the compositions tested, depending on the grammage, are small. Strongly evident are the differences in the burning rate of the composition containing 70% of fuel, which is probably related to oxygen balance and excess fuel.

## 5. Conclusions

This paper focused on investigating pyrotechnic delay compositions (PDCs) containing zinc as the fuel and KMnO_4_ as the oxidising agent. The compositions were found to be insensitive to friction and exhibited impact sensitivity in the range of 7.5–50 J. This allows them to be classified as relatively safe, as the measured values are similar to commercially used delay compositions.

The most important feature of the PDC element is the stable linear burning rate The results of the burning rate measurement showed that only compositions with a fuel content in the 50–60 wt. % range showed relatively high combustion stability, compared to the other PDCs. The other tested compositions showed insufficient repeatability of this parameter. The high pressure generated during combustion is not conducive to maintaining high stability during combustion. Because of these factors, unless additives are introduced, the application of the investigated formulations as PDCs will be limited, even despite their handling safety and environmental friendliness.

The results of pressure parameter investigations showed analogous trends as a function of fuel content and burning rate. Despite the decreasing oxygen balance, an increase of high pressure parameters is observed. This is due to the properties of the fuel used, which exhibits a low boiling point (approx. 907 °C).

DTA/TG analysis showed similar ignition temperatures for all tested samples. The lack of dependence on fuel/oxidising agent ratio indicates a complex multi-step ignition process, including the first step of oxidising agent decomposition, fuel melting, and second step of oxidising agent decomposition combined with actual ignition. The study also points to the important role of macroscopic fuel melting limiting the ignition and the diffusion of gaseous oxygen from oxidising agent particles.

SEM/EDS analysis showed that the morphology of the products differed remarkably as a function of composition. The analysis supports the postulated radical changes in the mechanism of combustion of delay compositions and corroborates the conclusions drawn from DTA/TG analyses.

In summary, these compositions are highly promising as ignition compositions for pyrotechnic igniters [35]. This is due to the fact that a mixture of hot gases, hot solid, and liquid particles are generated during their combustion, resulting in a very high ignition ability. Owing to the generation of solid/liquid products, due to their high heat capacity and good heat conduction coefficients, the impact of the initiating pulse generated by such a system can be extremely long and effective [36]. The fraction of condensed phase in the composition products limits the rate of heat transfer to the acceptor (e.g., rocket fuel, propellant) area.

Consequently, the investigated compositions are expected to be excellent replacements for typical ignition compositions, which contain heavy metal salts (e.g., the old RR-4 composition [37]) or compositions which generate toxic products (e.g., Mg/PTFE/Viton composition [38]).

## Figures and Tables

**Figure 1 materials-15-06406-f001:**
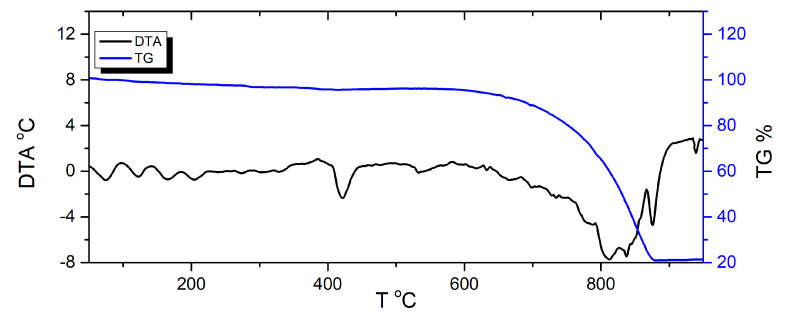
DTA and TG curves for KMnO_4_.

**Figure 2 materials-15-06406-f002:**
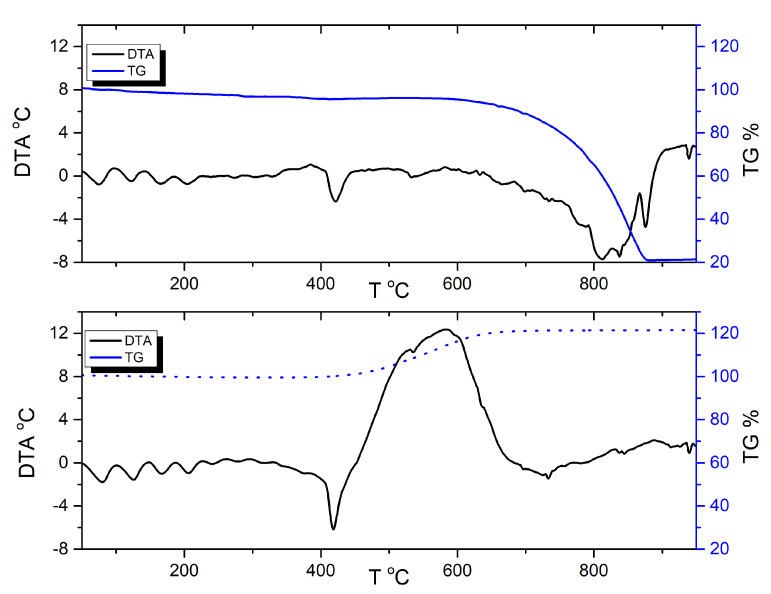
DTA and TG curves for Zn in argon (upper) and air (lower).

**Figure 3 materials-15-06406-f003:**
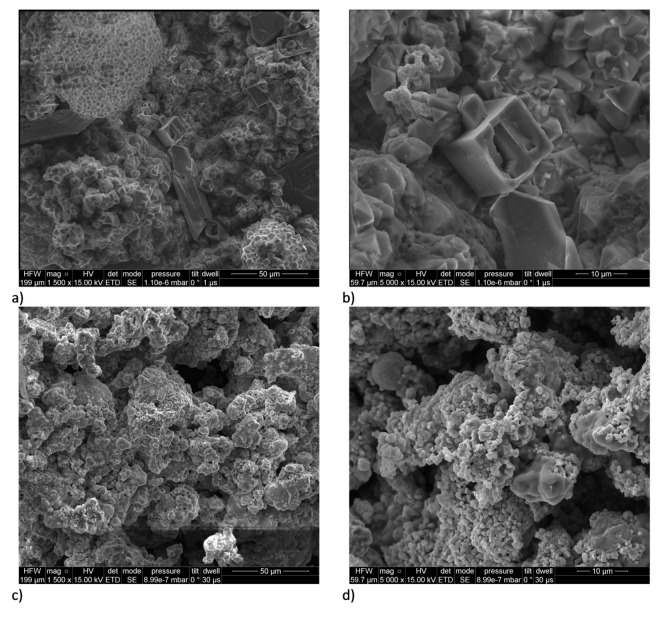
Result of SEM analysis for combustion products of samples Zn2 and Zn7. (**a**)—Zn2 with ×1500 magnification; (**b**)—Zn2 with ×5000 magnification; (**c**)—Zn7 with ×1500 magnification; (**d**)—Zn7 with ×5000 magnification.

**Figure 4 materials-15-06406-f004:**
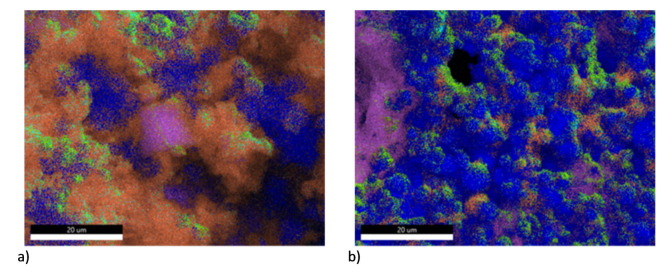
Result of EDS analysis for combustion products of samples Zn2 and Zn7. (**a**)—Zn2 combustion products; (**b**)—Zn7 combustion products.

**Table 1 materials-15-06406-t001:** Composition and sample naming scheme of investigated Zn/KMnO_4_ PDC formulations.

Composition Label	Zn1	Zn2	Zn3	Zn4	Zn5	Zn6	Zn7
Zn % wt.	35	45	50	55	60	65	70
KMnO_4_ % wt.	65	55	50	45	40	35	30

**Table 2 materials-15-06406-t002:** Impact sensitivty of investigated Zn/KMnO_4_ PDC formulations.

Composition Label	Zn1	Zn2	Zn3	Zn4	Zn5	Zn6	Zn7
Impact sensitivity [J]	>50	10	10	10	7.5	>50	>50

**Table 3 materials-15-06406-t003:** Grammage measurement.

ID of Pyrotechnic	Average Grammage	Standard Deviation	ID of Pyrotechnic	Average Grammage	Standard Deviation
Element	g/cm	g/cm	Element	g/cm	g/cm
L1L	0.176	0.018	LS1	0.214	0.013
L2L	0.177	0.023	LS2	0.217	0.016
L3L	0.176	0.018	L3S	0.226	0.022
L4L	0.177	0.023	L4S	0.211	0.015
L5L	0.180	0.018	L5S	0.228	0.014
L6L	0.177	0.011	L6S	0.236	0.006
L7L	0.185	0.015	L7S	0.226	0.014

**Table 4 materials-15-06406-t004:** Burning rate.

ID of Delay	Average Burning	Standard Deviation	ID of Delay	Average Burning	Standard Deviation
Element	Rate mm/s	mm/s	Element	Rate mm/s	mm/s
L3L	13.03	0.84	L3S	13.52	0.39
L4L	15.69	1.13	L4S	16.43	0.28
L5L	23.60	0.85	L5S	25.25	0.91
L6L	28.05	0.61	L6S	28.45	1.27
L7L	23.24	1.74	L7S	21.77	1.73

**Table 5 materials-15-06406-t005:** Pressure parameters.

Composition	Maximum Pressure kPa	Standard Deviation kPa	$\frac{\mathrm{dp}}{\mathrm{dt}}$ kPa/s	Standard Deviation kPa/s
Zn1	2.30 × 10^0^	1.64 × 10^0^	8.95 × 10^1^	1.11 × 10^1^
Zn2	2.19 × 10^1^	4.47 × 10^0^	8.91 × 10^1^	1.10 × 10^2^
Zn3	2.29 × 10^2^	1.58 × 10^1^	6.05 × 10^3^	4.66 × 10^2^
Zn4	4.13 × 10^2^	2.61 × 10^1^	2.13 × 10^3^	1.70 × 10^4^
Zn5	4.06 × 10^2^	2.73 × 10^1^	4.00× 10^4^	7.01 × 10^3^
Zn6	3.99 × 10^2^	1.13 × 10^1^	4.08× 10^4^	5.16 × 10^3^
Zn7	3.47 × 10^2^	1.95 × 10^1^	3.56× 10^4^	3.86 × 10^3^

**Table 6 materials-15-06406-t006:** Thermally-induced ignition temperatures of investigated Zn/KMnO_4_ PDC formulations.

Composition Label	Zn1	Zn2	Zn3	Zn4	Zn5	Zn6	Zn7
Preignition temperature [°C]	461.8	464.0	459.7	461.9	462.5	454.1	457.0
Ignition temperature [°C]	500.4	512.8	513.5	513.0	505.3	502.9	495.5

## Data Availability

Not applicable.

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
