# Peer review of "Investigation of Combustion of KMnO4/Zn Pyrotechnic Delay Composition"

_materials, 2022, doi:10.3390/ma15186406_

Round 1
Reviewer 1 Report
Please rewrite abstract and highlight the main findings in it.
Please check references 24-25.
The discussion section is very little and poor presented please add more description with evidence for results.
Too many subtitles specially in 2.material and methods. Please merge some of them together.
The quality of presentation for
last figures is very interesting please add some of it if possible in main text.
Author Response
Esteemed Sir / Madam,
We would like to thank you for your perusal of our manuscript and your comments, which have helped us improve its quality and clarity. Enclosed below, please find our responses (itallicised) to your comments (in bold):
"Please rewrite abstract and highlight the main findings in it."
The abstract has been rewritten to better highlight the findings of the manuscript.
"Please check references 24-25."
The references were corrected as indicated.
"The discussion section is very little and poor presented please add more description with evidence for results."
The discussion section has been significantly expanded, also including some of the subjects indicated by the other Reviewers.
"Too many subtitles specially in 2.material and methods. Please merge some of them together."
We have merged some subsections as indicated, however, any further merging might adversely affect the clarity of the manuscript.
"The quality of presentation for last figures is very interesting please add some of it if possible in main text."
The sections dedicated to SEM-EDS experiments have been expanded as indicated.
Reviewer 2 Report
The manuscript presents a pioneering experimental study on the use of Zn/KMnO4 as fuel/oxidant pair in a pyrotechnic delay device that is truly environmental friendly. Based on their carefully performed experiments, the authors conclude that in principle such a PDC formulation can be used in practical situations, albeit more refined research needs to be done both on the PDC and the detonator design, among other things to make the delay highly reliable. Despite the need of further work, I see the strength of the work in its innovativeness and careful experimentation. The authors’ explanations are meaningful while maintaining good style and readability. The work is definitely of interest to the general readership of Materials and therefore I recommend it for publishing subject to the suggestions below.
1) There are very few typographical errors, the easily visible one I noticed throughout the legends to Tables with experimental measurements is “standard derivation” instead of “standard deviation”.
2) There is no explicit mentioning in the text of the supporting material provided in the Appendix. I suggest the authors provide relevant note(s) in the text and to also to provide the Appendix in the form of an electronic supplementary material rather than being integral part of the article.
Author Response
Esteemed Sir / Madam,
We would like to thank you for your perusal of our manuscript, your kind words on our work and your comments, which have helped us improve its quality and clarity. Enclosed below, please find our responses (itallicised) to your comments (in bold):
"There are very few typographical errors, the easily visible one I noticed throughout the legends to Tables with experimental measurements is “standard derivation” instead of “standard deviation”."
The relevant fragment has been corrected as indicated.
"There is no explicit mentioning in the text of the supporting material provided in the Appendix. I suggest the authors provide relevant note(s) in the text and to also to provide the Appendix in the form of an electronic supplementary material rather than being integral part of the article.”
We have included mention of the figures given in the Appendix in the main text and have expanded the sections dedicated to SEM-EDS experiments.
Reviewer 3 Report
The manuscript sent for evaluation consists of 20 pages, including 11 pages of text with 6tables and 3 figures, 8 Appendixs and 1 page for references. References contain 35 bibliographic items.
After reading the text, in general, I assess the scientific quality of the publication as good. The article has a logical structure. An overview of the current state is the basis for creating an experiments. The test material is characterized, the research methods are explained, the results are presented in tables and graphs. The research is current.
I have following comments:
Page 2
Before line 79 is need add aim of article.
Page 4
Line 133 I think that term „ combustion velocity (Table 4) “is not correct. We use terms „burning rate“ .
Line 149-151 - please standardize the time (second) unit also with Table 4 and Table 5: ...10 K/min for all samples. The tests were performed in argon or in case of zinc also in air environment, with flow rate of 60 dm3/h in all cases.
Page 7
Figure 1 and Figure 2 need equal unit rating on the y-axis
Page 11: Conclusions
It is not correct content. Please, write results of your research.
Page 19
Figure A8. Please, make the same rate value „y“ axis DTA.
Author Response
Esteemed Sir / Madam,
We would like to thank you for your perusal of our manuscript and your comments, which have helped us improve its quality and clarity. Enclosed below, please find our responses (itallicised) to your comments (in bold):
"Before line 79 is need add aim of article."
The introduction section was modified and the relevant fragment was rewritten to better highlight the aim of the article.
"Line 133 I think that term „ combustion velocity (Table 4) “is not correct. We use terms „burning rate“ ."
The terminology was corrected across the entire manuscript, using burning rate as indicated.
"please standardize the time (second) unit also with Table 4 and Table 5: ...10 K/min for all samples. The tests were performed in argon or in case of zinc also in air environment, with flow rate of 60 dm3/h in all cases."
The units used in the manuscript have been standardised as indicated.
"Figure 1 and Figure 2 need equal unit rating on the y-axis"
“Figure A8. Please, make the same rate value „y“ axis DTA.”
The figures were corrected accordingly.
"It is not correct content. Please, write results of your research."
The conclusions section has been rephrased and modified as indicated.
Reviewer 4 Report
Dear Authors,
Following are some observations:
1.Add some latest references
2. Author should add the property table of Zn and KMnO4 and mixture
3. Need discussion about stability and compatibility of Zn and KMnO4 mixture
4. How combustion flame speed is measured? Add effect of flame velocity on combustion.
5. When combustion takes place what is fuel and O2/Air ratio, if possible include in discussion.
Author Response
Esteemed Sir / Madam,
We would like to thank you for your perusal of our manuscript and your comments, which have helped us improve its quality and clarity. Enclosed below, please find our responses (itallicised) to your comments (in bold):
"Add some latest references"
Multiple recent works were additionally cited in the manuscript. Unfortunately remarkably few recent works are directly related to the development of this class of pyrotechnic delay compositions.
"Author should add the property table of Zn and KMnO4 and mixture "
The most relevant properties of the compositions have been gathered in Tables 1 & 2, whereas the individual components have been more thoroughly described in Section 2.1.
"Need discussion about stability and compatibility of Zn and KMnO4 mixture"
The discussion section was modified accordingly to include this subject.
"How combustion flame speed is measured? Add effect of flame velocity on combustion.”
The relevant measurement details have been added to the experimental section. The correlation of flame velocity and combustion has been briefly outlined in the discussion section.
"When combustion takes place what is fuel and O2/Air ratio, if possible include in discussion."
In the DTA/TG experiments, the samples are heated under gas flow during their decomposition. As the fule content changes during this process and the air is constantly being exchanged, it is impossible to even estimate the fuel / air ratio during those experiments.